# Automatic brain lesion segmentation on standard magnetic resonance images: a scoping review

Emilia Gryska ,[1] Justin Schneiderman ,[2] Isabella Björkman-Burtscher ,[3] Rolf A Heckemann [1]

► Prepublication history and additional materials for this paper is available online. To view these files, please visit the journal online (http://dx.doi.org/10.1136/bmjopen-2020-042660).

¹Medical Radiation Sciences, Goteborgs universitet Institutionen for kliniska vetenskaper, Goteborg, Sweden
²Sektionen för klinisk neurovetenskap, Goteborgs Universitet Institutionen for Neurovetenskap och fysiologi, Goteborg, Sweden
³Radiology, Göteborgs universitet Institutionen för kliniska vetenskaper, Goteborg, Västra Götaland, Sweden

**Correspondence to**
Emilia Gryska;
emilia.gryska@gu.se

## ABSTRACT

**Objectives** Medical image analysis practices face challenges that can potentially be addressed with algorithm-based segmentation tools. In this study, we map the field of automatic MR brain lesion segmentation to understand the clinical applicability of prevalent methods and study designs, as well as challenges and limitations in the field.

**Design** Scoping review.

**Setting** Three databases (PubMed, IEEE Xplore and Scopus) were searched with tailored queries. Studies were included based on predefined criteria. Emerging themes during consecutive title, abstract, methods and whole-text screening were identified. The full-text analysis focused on materials, preprocessing, performance evaluation and comparison.

**Results** Out of 2990 unique articles identified through the search, 441 articles met the eligibility criteria, with an estimated growth rate of 10% per year. We present a general overview and trends in the field with regard to publication sources, segmentation principles used and types of lesions. Algorithms are predominantly evaluated by measuring the agreement of segmentation results with a trusted reference. Few articles describe measures of clinical validity.

**Conclusions** The observed reporting practices leave room for improvement with a view to studying replication, method comparison and clinical applicability. To promote this improvement, we propose a list of recommendations for future studies in the field.

## INTRODUCTION

MRI has become an integral part of diagnostics to detect, differentiate and characterise brain lesions. Properties that determine this success are safety, high tissue contrast and sensitivity to abnormality. Conditions that can lead to brain lesions include traumatic brain injury, vascular disease (including ischaemic and haemorrhagic stroke), neoplasms, autoimmune disorders, infection, degenerative diseases, congenital conditions and systemic diseases with secondary effects on the central nervous system. In some of these conditions, accurate estimation of lesion size and its progression may be essential to treatment

---

**Strengths and limitations of this study**

► This is the first overarching review of MR-based automatic brain lesion segmentation methods without restriction to a particular lesion type.
► The study was conducted following a previously published protocol.
► We consulted practitioners to ensure relevance of our work for clinical practice.
► The rigorous study design restricted our ability to expand on emerging themes, but did not entirely prevent epistemic drift.
► The restriction to a one-pass full-text analysis entailed a superficial level of interpretation—in hindsight a deeper reading would have been desirable.

---

planning, disease monitoring and/or treatment evaluation.

Visual image interpretation is still the most common and accepted way to analyse clinical images. In some cases, the visual examination must be extended to include lesion delineation. For example, the boundaries of a tumour and structures at risk need to be localised accurately for radiation therapy planning. Such delineation depends on a skilled rater, is time consuming, and is characterised by high variability between raters and high variability between repeated delineations on the same images by the same rater (ie, low objectivity and reliability, respectively). In other clinical examinations, surrogate metrics, which can be based on the largest perpendicular diameters[1 2] or visual scoring,[3] are used for estimating the size of a lesion rapidly. Such visual or area-based lesion volume estimates may not be accurate enough.[4]

Developments in magnetic resonance (MR) image acquisition procedures resulted in high-quality and complex images that capture a variety of structural and functional phenomena. On one hand, the images carry more information about the lesions, thereby enabling more accurate diagnosis. On the

other hand, each image requires more analysis time. This reduces the rate at which radiologists can analyse images. The increasing accessibility to MR scanners furthermore results in more images being acquired, further straining image processing capacity and potentially diminishing the quality of interpretation.[5][6]

To summarise, three major issues with current clinical practices for image analysis are: subjective and time-consuming manual segmentation, limited accuracy of the surrogate volume estimates, and increasing information content and number of images to process.

Automatic lesion segmentation algorithms promise to alleviate these issues through fast and consistent lesion delineation that scales with demand. Quantitative image analysis using automatic lesion segmentation further has the potential to increase diagnostic and prognostic accuracy of lesion examination by providing radiologists with prompt and explicit information about a lesion.

Several reviews on automated brain lesion segmentation methods have previously been published. Most of these outline and analyse commonly used segmentation algorithms or general segmentation principles.[7–16] The perspective of the clinical applicability of automated segmentation algorithms has been explored by Garcia-Lorenzo et al[10] and Bauer et al.[7] In both reviews, the authors recommend improvements concerning study designs and formulation of research questions (RQs). The authors stress the importance of validating the robustness of the segmentation algorithms under variation stemming from three causes: differences between scanners and acquisition protocols, natural variability in normal anatomy and lesion appearance, and artefacts. Moreover, Bauer et al[7] expressed the need for better communication between researchers developing segmentation methods and clinical radiologists who are the intended users of these methods.

A comprehensive review of automatic segmentation methods without restriction to lesion type is still lacking today, perhaps because of the challenge associated with the large and growing literature. We therefore present a scoping review of such methods and, given the rapidly growing extent of the literature, provide a timely account of the field that may not be feasible in the future. To ensure rigour, reproducibility and comprehensiveness, we adopted scoping review methods proposed earlier.[17–21]

Furthermore, there is a disconnection between developers and users of automatic brain lesion segmentation methods.[7] We therefore aim to examine clinical relevance of the research conducted in the field and seek to understand how the most prevalent methods (segmentation algorithms) and study designs (how they are validated) reflect the clinical applicability of methods described in the reviewed articles. Moreover, we aim to identify issues, limitations, and grand challenges of the field, and suggest actions to bridge the gap between research and clinical practice.

## METHODS AND DESIGN

This study has been conducted according to a previously published protocol[22] and based on scoping review methods proposed by Arksey and O'Malley[17] and further developed by Levac et al and Colquhoun et al.[18][19] We also incorporate relevant parts of the Preferred Reporting Items for Systematic Reviews andMeta-Analyses (PRISMA)[20] and PRISMA extension for Scoping Reviews[21] protocol for this type of review. Here, we provide a summary of the protocol as well as an account of changes made, along with their respective rationale.

### Stage 1: identifying RQs

The following RQs were posed in the protocol[22]:

1. Which common image processing steps are necessary for automatic brain lesion segmentation on MR images?
2. Which mathematical and computational theories are most commonly applied in which types of brain lesions?
3. What is the efficacy of existing implementations?
4. What are the limitations of those methods and issues that should be addressed in future studies to develop a tool that is suitable for clinical use?
5. What are the most commonly used MR data sets that provide reference lesion segmentation and/or diagnostic classification?

While getting familiar with the abstracts of the sample, we questioned the relevance of RQs 1 and 2. We recognised the need for consultation at this early stage of the project as opposed to consultation on the findings as originally proposed by Arksey and O'Malley.[17] The RQs do not reflect fully the potential of our study to address issues critical to advancing the field and bringing a benefit to the community. The consultation was conducted as semistructured interviews with five clinicians who have experience in brain image analysis. We interviewed two neurosurgeons, an oncologist, a radiation oncologist and a neuroradiologist. The interviews were structured to elicit understanding of their daily workflow, of how they use and analyse images, and of their opinions on automating the process of brain lesion segmentation.The initial screening of the sample abstracts and, in parallel, the consultations directed our analysis towards examining the clinical relevance of the prevalent methods and study designs.

### Stage 2: identifying relevant articles

For the purpose of this study, we define an article as an individual published item that was found according to the presented search strategy and that meets the inclusion criteria of this review.

Eligibility criteria were unchanged from those defined in the protocol: articles evaluating an automatic brain lesion segmentation method applied to MR images acquired for human brain lesion inspection, and articles evaluating the performance of the method by comparison with a reference segmentation were suitable for the study.

The initial search strategy remained mostly unchanged. Three databases (PubMed, IEEE Xplore and Scopus) were

queried. Initially, a broad search phrase on automatic brain lesion segmentation methods on MRI was used to generate a first sample for each database. Controlled vocabulary tags were extracted from the articles and arranged in descending order of frequency. From this list, tags were selected to refine and customise the search phrase for each database. These refined and customised search phrases were used to retrieve the sample for our study. The search was conducted on 4 November 2018.

The relevant articles were identified through hierarchical screening at four levels with given aims and objectives:

1. Title level: exclusion of clearly ineligible articles.
2. Abstract level: refinement of inclusion and exclusion criteria based on literature content; identification of 100 eligible articles by reading of randomly selected abstracts out of a sample of 1359; identification of themes to guide the study selection process (stage 3).
3. Methods level: identification of two core concepts: algorithm validation and applicability of an automatic segmentation method for clinical parameter prediction; exclusion of studies using manual or semiautomatic segmentation, not stating the source of the tested images/patients, only using synthetic images, conducing the algorithm validation on fewer than 10 unique scans/images, or not providing results of the algorithm's performance.
4. Full paper screening level (see stage 3).

The same hierarchical screening procedure was applied to references of the eligible sample also screening for duplicates and excluding conference proceedings that were later published in journal articles if the patient cohort was similar in both articles. Screening was discontinued on finding an exclusion criterion.

## Stage 3: study selection

In the full-text screening, we applied the above criteria again and excluded papers that:

▶ Did not perform segmentation or did not provide segmentation evaluation outcome measures (wrong outcome).
▶ Proposed methods that require user interaction.
▶ Did not provide information about the gold standard or a reference measurement.
▶ Did not provide information about the origin of the images or used only synthetic images.
▶ Did not provide the number of images used for evaluation.
▶ Evaluated the method on fewer than 10 unique scans.

## Stage 4: data charting

The data extracted were charted in a spreadsheet (online supplemental material) with five categories of variables:

1. Bibliographic category (first authors' name, article title, journal or conference name, and year of publication).
2. Segmentation category (preprocessing procedures, algorithms and computational theories, if lesion classification used, required or allowed input modalities, processing time of the method and availability of the software).
3. Validation category (process of obtaining reference segmentations, number of raters or observers, and evaluation metrics).
4. Study cohort category comprised patients' diagnoses, lesion type, sources and number of images).
5. General comments category (information that did not fit in other categories, but was considered relevant) stage 5: collating, summarising, and reporting the data.

## Stage 5: collating, summarising and reporting the data

The collected data are disseminated through numerical analysis of bibliographic information overview, trends in the field and study design characteristics. We provide the account of both prevalent study design choices and less common ones that are relevant to the clinical applicability assessment. Study design features are grouped into materials, preprocessing, performance evaluation and performance comparison.

## Patient and public involvement

No patients nor members of the public were involved in this research project.

## RESULTS

The results of the scoping review include answers to RQs 2, 4 and 5. The answers to questions 1 and 5 are directly presented in the results. The answer to question 4 is formulated in the discussion based on the presented findings. In the discussion, we also justify why questions 1 and 3 could not be explicitly answered.

### Overview

The number of papers included at each stage of the review is shown in figure 1. Among 2500 articles returned by the searches and 490 titles identified through reference screening, we included 441 articles in the review. Among the eligible papers, 255/441 were published in journals[23–277] and 184/441 in conference proceedings or conference workshops,[278–460] and 2/441 articles were uploaded as preprints.[461 462] The following journals or conferences occurred most frequently as the publication source: International Conference on Medical Image Computing and Computer-Assisted Intervention (66 articles), *NeuroImage* (20 articles), *IEEE Transactions on Medical Imaging* (20 articles), *SPIE Medical Imaging Conference* (18 articles), *NeuroImage: Clinical* (15 articles), *IEEE International Symposium on Biomedical Imaging* (13 articles) and *PloS One* (12 articles).

Figure 2 shows the distribution of articles that propose a method to segment a particular brain lesion type. The dominance of methods segmenting brain tumours is evident from the data.

### Trends

Substantial growth of the field was evident from the number of articles published annually (figure 3). Three

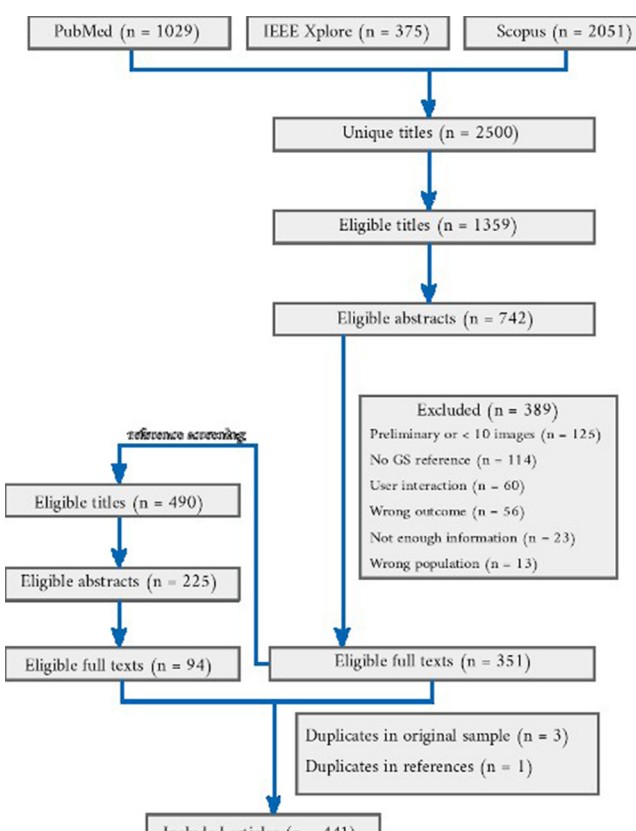

**Figure 1** Flow chart of the article selection process from the result of querying three databases. For each stage, the number of articles selected is shown and numbers of excluded articles and reasons are given.

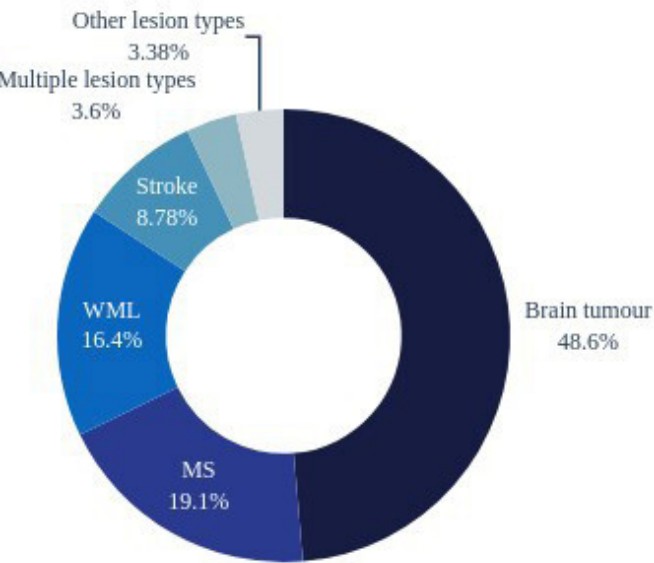

**Figure 2** The number of algorithms described in the included articles developed for and validated on particular lesion type. The distinction is made between multiple sclerosis (MS) lesions and other white matter lesions (WML) following the similar distinction present in the reviewed articles and lesion segmentation challenges. 'Multiple lesion types' refers to algorithms that were evaluated on more than one type of lesion. 'Other lesion types' types of lesions included focal cortical dysplasia, metastasis, traumatic brain injury, cerebral palsy, abscess and necrosis.

distinct periods with an increase in the number of published articles compared with preceding years were noticeable. During these years, an increase is also evident in the number of articles addressing segmentation of lesion types corresponding to the challenges taking place. The first wave corresponds to the *3D Segmentation in the Clinic: A Grand Challenge II: MS Lesion Segmentation*.[463] The second wave started in 2012 coinciding with the advent of *The Multimodal Brain Tumor Image Segmentation Benchmark (BraTS)*.[464] The third wave started in 2015 when three brain lesion segmentation challenges took place: BraTS, the *Longitudinal MS Lesion Segmentation Challenge*,[465] and the *Ischemic Stroke Lesion Segmentation challenge (ISLES)*.[466]

### Image sources and databases
The distribution of the image sources is presented in table 1. Most commonly, the image data were collected from non-public sources (254 articles). Publicly available data with reference segmentations were used in 217 articles. The most popular database was *BraTS*[464] (157 articles) followed by the *3D Segmentation in the Clinic: A Grand Challenge II: MS Lesion Segmentation*[463] and *ISLES* (17 articles).[466] 10 articles rely on publicly available sources without reference segmentations. Forty-one articles validate the method on both publicly available data sets with reference segmentations and non-public data.

The distribution of the cohort sizes used to validate the proposed methods (figure 4) shows that more than half of the studies described in included articles test methods on 50 or fewer individuals' brain images (260/441). Among the eligible articles, 280/441 describe a method validated on images obtained from more than one scanner.

### Sequences
The majority of the proposed methods operated on multi-sequence scans as inputs (307/441) while 6 accepted both

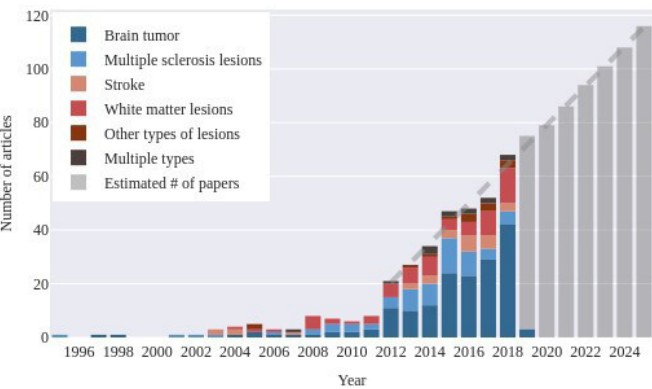

**Figure 3** The number of articles published per year in the eligible sample with an indication of the number of articles that presented a segmentation method for a particular lesion type. We fit a linear function to the number of papers published each year from 2012 to estimate the development of the field in the upcoming years.

**Table 1** Distribution of image sources used for algorithm validation

| Non-public data sources | Publicly available data with reference segmentations | | | | Publicly available data without reference segmentations |
| | BraTS[464] | Grand challenge II: MS lesion segmentation[463] | ISLES[466] | Other | |
| --- | --- | --- | --- | --- | --- |
| 254 | 157 | 31 | 17 | 12 | 10 |

BraTS, Multimodal Brain Tumor Image Segmentation Benchmark; ISLES, Ischemic Stroke Lesion Segmentation challenge.

single-sequence and multisequence input. Seventy-nine methods were built to evaluate single-sequence input and 22 did not specify the input modality. Articles that compared or evaluated a previously proposed method were not included in the analysis (27/441).

## Image processing steps, algorithms and computational theories

### Preprocessing procedures

The most commonly applied preprocessing procedures and tools (disregarding publicly available databases for this purpose) are shown in table 2. Although authors commonly document individual steps and corresponding algorithms, information on whether these algorithms are integrated in the segmentation tool is rarely available.[90 195] Fifteen articles mention visual supervision,[93 114 133] semiautomatic preprocessing,[186 226 267 328 405] manual correction,[155 171 209 273] error or failure of preprocessing (usually at the brain extraction step).[135 136 160]

Five methods proposed in the reviewed articles use minimal[268 400] or no preprocessing,[163 164 220] and in each case this is presented as an advantage of the proposed algorithm.

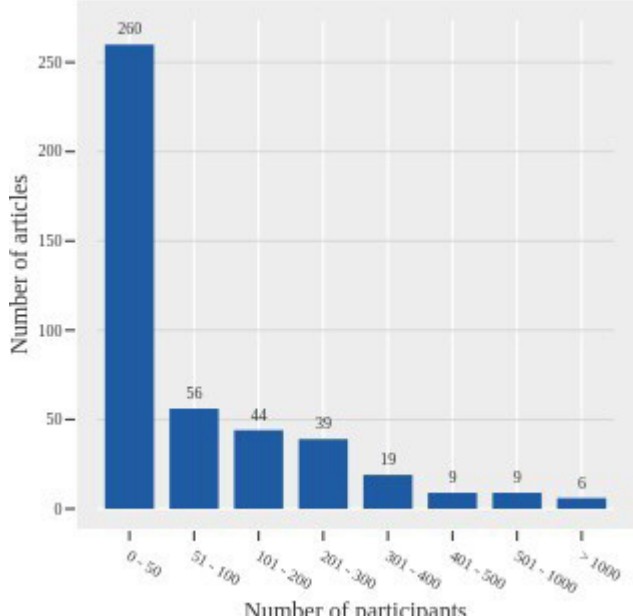

**Figure 4** The number of patients (n) whose images were used to validate the methods in the eligible articles.

### Algorithms and computational theories

Among the most commonly used algorithms or models in the segmentation methods were machine learning (decision trees, mixture models, fuzzy clustering, support vector machines, expectation maximisation) and in particular deep-learning algorithms (artificial neural networks) (figure 5). The analysis accounted for the same methods published in separate articles that deal with either different lesion types or similar lesion type with different image sources.

### Performance evaluation

Comparison with the reference by overlap measures was by far the most commonly used criterion to evaluate the accuracy of the automatic segmentation. Among articles that specify how the reference segmentations had been generated[265] in publicly non-available data sets, most specified manual segmentation by one rater[85] or two raters.[61] More than two raters contributed in 38 articles, and 33 articles used semiautomatic procedures for constructing the reference. When multiple raters delineated the region of interest, the reference segmentation was obtained through consensus of the raters or majority voting. Forty-five articles have also included inter-rater and intrarater variability evaluation on data sets used in their studies. The most frequently applied measures were the Dice coefficient,[330] sensitivity, also expressed as true positive ratio TPR, or overlap fraction,[227] positive predictive value,[102] Jaccard coefficient,[58] and specificity.[82] Volumetric measures, such as volume difference, error

**Table 2** Prevalent image preprocessing steps in 374/414 articles (not specified in 67 articles). FSL - FMRIB Software Library; BET - Brain Extraction Tool; SPM - Statistical Parametric Mapping.

| Procedure | Common tool/approach | N (of 374) |
| --- | --- | --- |
| Intensity normalisation | Histogram matching, intensity scaling | 224 |
| Bias field (field inhomogeneity) correction | N3, N4[476 477] | 192 |
| Brain extraction | FSL BET[478] | 190 |
| Image (co-)registration | rigid/affine; SPM, FSL[479 480] | 179 |
| Denoising | anisotropic diffusion filtering | 60 |
| None or minimal | | 5 |

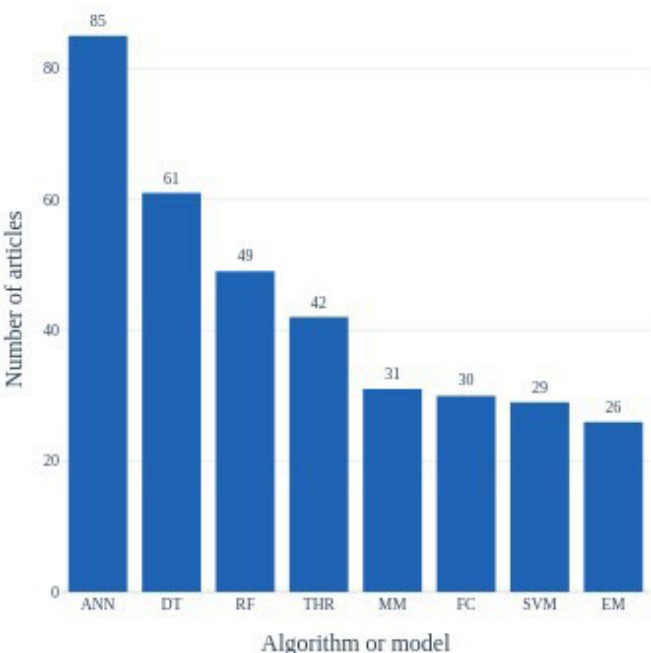

**Figure 5** Distribution of the most common algorithms and models implemented in the lesion segmentation methods proposed by the articles eligible for the review. ANN - artificial neural networks; DT - decision trees; EM - expectation maximisation; FC - fuzzy clustering; MM - mixture models; RF - random fields; SVM - support vector machines; THR - thresholding.

or correlation between the automatic and reference segmentation were used in 133 articles, and distance-based measures (such as Hausdorff distance, mean absolute distance, surface distance) in 41.

In addition to segmentation accuracy, reliability of a method was estimated in some of the articles. Reliability was directly evaluated through a test–retest procedure, where participants were scanned twice at a short interval, with repositioning between the scans in five articles.[127 177 196 321 347] In other articles, indirect approaches were described where consistency of longitudinal data with clinical findings of stable or progressive disease was used as a proxy for reliability.[38 270 454]

### Performance comparison

Maldjian *et al*,[133] Lesjak *et al*[150] and de Sitter *et al*[224] examined previously proposed methods and compared the results they obtained in their studies to the performance reported in the original papers. All three articles reported poorer independent testing results. The authors pointed out a lower lesion load of the studied population than in the original papers that accounted for reduced scores in the replication attempts. In their paper, de Sitter *et al*[224] strongly called for improvements to automatic lesion segmentation before their introduction to routine clinical use.

Of 441 articles, 233 included information regarding both the processing time (from less than 1 s to 7 hours) and computational system requirements used for

segmentation. In 56 articles, we found information about only one of the two parameters; however, such information is incomplete and cannot be used to estimate how well a method will perform on different hardware.

Only 24 of the included articles reported on methods that were made publicly available to download.[43 50 53 67 76 84 90–92 126 134 138 144 149 154 170 173 178 190 195 196 199 228 230 426] One was available on request.[111] Two articles promised future availability of the methods proposed[41 193] but they were not available as of 4 May 2020.

### Reporting recommendations

Based on our findings, we developed a checklist with reporting recommendations (table 3).

## DISCUSSION

In this paper, we present a scoping study of automatic brain lesion segmentation on MR images based on rigorous literature review methodology.[17] This is the first review that takes all methods into account, independent of specialisation towards lesions of a particular aetiology. The key findings from our analysis of 441 articles are (1) a rapid increase of interest in the field, (2) a plethora of proposed methods contrasted with a dearth of open documentation and available software, and (3) high prevalence of problematic reporting practices that restrict the ability of independent researchers to replicate reported results and conduct method comparisons.

### Variable design of automatic segmentation methods

#### RQ1: Which common image processing steps are necessary for automatic brain lesion segmentation on MR images?

Image preprocessing emerged as a pervasive step in the processing chain for lesion segmentation. From the collected data we identified procedures that are commonly agreed to constitute preprocessing: skull stripping, image coregistration, bias field correction and intensity normalisation. Some algorithms rely on additional preprocessing, such as tissue classification. The procedures vary considerably between methods, and a lack of a universal distinction between segmentation steps and preprocessing steps became apparent during the analysis.

For a clinically suitable method, a distinction between preprocessing and segmentation steps may not be necessary since the final segmentation will rely on a whole processing chain (including the preprocessing) applied to a raw image. Validation of the whole chain, and assessing the impact of each step on the outcome, however, are desirable in method evaluation studies. Unfortunately, authors rarely state whether preprocessing is integrated with the segmentation algorithm and to what extent it relies on user input. Without such information, namely a list of all the steps that are performed on a raw image, the study cannot be replicated. This implies that the findings

**Table 3** Proposed reporting items for automatic brain lesion segmentation studies

| | | |
|---|---|---|
| Technical validation | Method | List of processing steps necessary to apply to a raw image |
| | | Computational system parameters |
| | | Computation time |
| | | Open documentation of the algorithm |
| | Reference segmentation | Number of raters |
| | | Raters' training/experience |
| | | Method of segmentation |
| | | Method of consolidation (if multiple raters) |
| | Validation | List of validation metrics used |
| | | Number of images (split into training/validation/testing if applicable) |
| | | Input sequences |
| | | Number of scanners used to acquire images |
| | | Acquisition parameters |
| | | Number of time points and intervals for longitudinal data |
| | Results | Mean, median, SD for each validation metric |
| | | Number of failed cases (if applicable) |
| Preclinical validation | Patient information | Diagnosis and level of verification for example, clinical follow-up, tissue sampling, autopsy, etc. |
| | | Clinical presentation |
| | | Administered treatments (if applicable) |
| | Clinical task | Explicit definition of the clinical task for which the algorithm is applied (eg, lesion growth estimation, treatment evaluation, radiotherapy planning). |
| | User validation | List of optimisation metrics used for the clinical task (if applicable). |
| | | User's method of evaluating the outcome. |
| Clinical validation | Information and storage system compatibility | Compatibility with picture archiving, communication, and storage systems |
| | Regulatory approval | Compliance with the regulatory approvals for software as a medical device. |

are of no help in assessing the potential clinical validity of the proposed method implementation.

Another implementation issue arises from the necessity of visual supervision or manual corrections of the preprocessing that was indicated in 15 articles. Any requirement of user interaction at any point in the processing chain entails disadvantages: it impedes implementation by increasing the complexity of the technical integration, and, once the tool is implemented, permanently burdens staff with an additional task. Any potential benefit of user interaction (eg, increased robustness) has to be weighed carefully against these costs.

### RQ2: Which mathematical and computational theories are most commonly applied in which types of brain lesions?

The most prevalent methods used were artificial neural networks. The finding is not surprising given the popularity and remarkable performance of deep learning algorithms particularly in image processing applications. In the absence of widely used and agreed-upon criteria for performance evaluation, we have abstained from attempting to rank methods or make specific recommendations.

## Validation process and efficacy of automatic segmentation methods

### RQ2: What is the efficacy of existing implementations?

We found that in view of current practices, the question cannot be answered: authors present assessments that are predicated on their own needs and biases, and there is no established standard that enables fair method comparison. Public challenges have been set up in an effort to address this problem, but they have only been partially successful, as illustrated in the following section. Three obstacles to fair comparative method assessments are paramount: the principal lack of ground truth in in vivo imaging, the enormous parameter space of acquisition settings that leads to variable feature presentations, and the lack of objectivity (as shown by inter-rater and intrarater variability[467 468]) of reference segmentations. More effort should thus be devoted to increasing the informativeness of the validation step. This can be achieved by painstakingly reporting the number of annotators, their experience, how the delineations were fused, and acquisition details that could conceivably have an impact. According to Gibson et al,[469] such information can be used to calculate the statistical power of segmentation studies with respect to the number of reference images.

To answer the question we posed regarding the efficacy, we recorded the values of the comparison measures from each article. The information, however, cannot be meaningfully synthesised or compared and was not presented in the results section for this reason. Our investigation did not result in a direct answer to the question, but revealed a paucity of standard comparison procedures.

Inability to answer the posed question points to further issues on which we reflect from the perspective of clinical suitability of the proposed methods. The medical image analysis procedures commonly used in clinical diagnostics today are strongly operator dependent. They also scale poorly to the growing workload that results from the increasing number of modalities,[470] the increasing

accessibility of imaging scanners,[471] and the increasing amount of image information due to technical advances that achieve enhanced spatial and contrast resolution.[472] Automatic image analysis methods in general, and among them automatic lesion segmentation methods promise to alleviate some of this pressure. Other requisite characteristics that need to be assessed to evaluate the clinical applicability of a tool are reliability, robustness, and generalisability of its findings.

Reliability of an automatic segmentation algorithm seems to be assumed since the decision whether a given voxel is a lesion or not is made based on well-defined rules. However, the characteristic is rarely tested in method validation studies. Only five articles in our sample included a test–retest evaluation with patient repositioning between the scans. Such testing yields crucial reliability data against which longitudinal variations of lesion measures need to be compared with distinguish actual lesion change from other sources of variability.

Generalisability of a method is tested by processing images acquired from different scanners on a sufficient number of images from populations that are at least as heterogeneous as the population expected to be examined clinically. While more than half of the articles (280/441) used images coming from more than one scanner, validation on fewer than 50 images were equally prevalent (260/441). Such small sample sizes are insufficient for clinical validation, even if the images originate from multiple scanners. Using images acquired from multiple sources and validating a method on a large number of images reduces a method's bias towards the data set or sets on which a given method was originally developed.

To us, one of the most surprising findings was how few methods are publicly available. Authors who share their software enable independent testing in unanticipated conditions and with various cohorts. Finally, to evaluate clinical relevance of the results of automatic segmentation, a method should be tested in genuine clinical scenarios for a well-defined task, such as monitoring disease progression, treatment response evaluation or radiotherapy planning. Each of these tasks may require a different level of error margin and validation standard and the relevance of the results must be evaluated by clinicians. Various metrics may be used to either optimise the method or evaluate its accuracy for a particular clinical question, lesion type and size.

Many papers claim usefulness of their algorithm for particular clinical tasks, but do not test the acceptability of the results for the intended purpose.

### Limitations and issues of the proposed methods, and grand challenges of the field

#### RQ4 (a) What are the limitations of those methods (…)?

Due to the inherent reference, image, and lesion variability, a meaningful comparison of algorithms evaluated independently on separate data sets to establish which method performs better is challenging. One solution to the problem has been proposed in the form of segmentation challenges. The challenges have become a quasi-standard for comparing brain lesion segmentation algorithms. The setup, however, comes with certain shortcomings with respect to evaluating clinical applicability of the evaluated algorithms. As mentioned, evaluating the whole chain of processing steps is especially important for validation of a clinical tool. In a challenge set-up, the algorithms are tested on partially preprocessed images. Even when the individual steps are described in detail, the impact of the preprocessing on the segmentation algorithms outcome is unknown. Validation of the processing chain becomes questionable if a step is changed, either based on the users' subjective judgement or due to implementation of a different procedure for a particular step. According to *ISLES*[466] and several articles in our sample, the skull-stripping step may need to be supervised and manually corrected. The choice of interpolation method is yet another issue that can influence the outcome of an automatic segmentation algorithm, especially in tasks requiring high accuracy.

Another aspect of the challenges is the prestige and publicity for the authors and their method after proposing a winning segmentation algorithm. The final rank is calculated by the organisers who evaluate the methods on hidden data sets. Organisers effectively take the role of independent arbiters. Participation in challenges and postcontest use of the challenge data as reference material indicate that organisers are generally trusted in this role. Maier-Hein *et al* suggest, however, that the results of such competitions should be considered carefully.[473] The authors point to several issues that have a significant impact on the final ranking of evaluated methods. The first problem they report is lack of thorough reporting of relevant information that is essential for result interpretation. Moreover, it turns out that changes in metrics and aggregation methods for the scores of individual test cases alter the ranking of evaluated methods. A similar effect on the ranking was observed when the reference segmentations were exchanged against those of another rater. The final rank in such a competition depends also on the test data and on how missing data is handled.

These concerns cast doubt on the validity of testing clinical applicability of automatic brain lesion segmentation methods in a challenge setup. Even if the challenge databases contain images from multiple scanners, it cannot be regarded as generalisable if it has not been tested on raw images (reconstructed, but not otherwise processed) as produced by the scanning equipment. In our sample, only 10% of the articles explicitly report on such evaluation. While obtaining independent data sets is expensive, it is a crucial step on the way of creating a clinically applicable automatic lesion segmentation tool.

Ultimately, it is the target users who need to decide whether a given validation standard and resultant performance confidence are sufficient to apply a tool to answer a given clinical question. With guidance from support system developers, as well as transparent and thorough

reporting of the processing steps and validation procedures, the users may consider a given tool trustworthy. Trustworthiness has been reported as a crucial factor in developing a clinically usable support tool.[474] The present scoping review does not show what properties and features clinicians need to develop trust in a tool. Still, this is an important question that we will seek to address in future work.

We conducted an exploratory study to begin to address the issue.[475] The aim was to learn how considering radiologists' competent input can improve the design and validation procedures of automatic brain lesion segmentation methods with a view to increasing trust and trustworthiness. Our findings corroborate previous findings.[474] We found that two crucial characteristics that influence clinicians' trust in a tool are the provision of an error margin with any quantitative measure, and consideration of the varying need for accuracy, depending on the diagnostic task.

### Relevance for research and clinical practice

Noteworthy trends also emerged from the data regarding the prevalence of methods developed for segmentation of a particular lesion type, as well as the advent of segmentation challenges focussing on corresponding lesion types. We observed that the majority of the methods described in our sample have been developed for the purpose of brain tumour segmentation. We also note consistent growth of the number of images in the database underlying the *BraTS* challenge, which has been organised annually since 2012. Stroke lesion segmentation methods have not proliferated to the same extent, despite similar competitions having been organised (yearly *ISLES* challenges from 2015 to 2018). A distinct increase in the number of publications proposing white-matter lesion segmentation algorithms was observed when segmentation challenges of this type of lesion were organised.

Some compelling questions follow this observation, for example, whether some tasks are more difficult to solve algorithmically, or whether these trends reflect on clinical usefulness of developing segmentation algorithms for a particular lesion type. Even though the questions cannot be directly answered by our findings, they point to important issues that need to be explored to better understand the relevance of different lesion segmentation algorithms and implications of the trends for clinical practice.

Another finding that potentially contributes to the gap between research and clinical practice is the widespread use of deep-learning algorithms for the lesion segmentation task in the research setting. The practical efficiency of the algorithms depends on the availability of graphical processing units. Implementing a deep-learning based algorithm in a clinical setting requires dedicating resources for purchasing suitable hardware and integrating it with radiological workflow, information and storage systems.

A key factor in developing a clinical tool is obtaining regulatory approval for diagnostic use. None of the articles in our sample mentioned this requirement, possibly because authors do not consider it relevant at the stage of development when publication occurs.

It appears that despite claims of clinical relevance made by many authors, development of brain lesion segmentation methods happens predominantly in an academic space, where technological challenges matter most and implementation hurdles are not explicitly considered. To increase the relevance and societal benefit of method development in the field, it will be necessary for developers to widen their perspective to include the critical path towards clinical implementation, on which users' demands and regulatory requirements have to be met.

### Challenges and limitations of the study

The biggest challenges we had to address in this study were designing data charting categories, extracting the relevant information regarding both inclusion and exclusion criteria, and the actual charting of the data. Arksey and O'Malley[17] recommend deciding the inclusion criteria after becoming familiar with the identified relevant studies. For us, this was impossible due to the large size of the raw sample of original search results, along with the variability of both the type and the level of detail of reported information.

Our inclusion and exclusion criteria were defined based on a pilot analysis of articles selected randomly from the raw sample. These criteria were then applied to the full raw sample in hierarchical fashion. A major impediment at this stage was the inconsistent way the information is organised and presented in the articles. In the case of conference papers, the authors typically have to adhere to page limits, so certain compromises are inevitable. When it comes to journal publications, the methodology of a study should be thoroughly reported for the sake of reproducibility, but also to enable data extraction for systematic reviews.

Another challenge we faced and a limitation of the study comes from the mentioned incoherent way of reporting studies and a challenge to derive strict definitions of the inclusion and exclusion criteria. Often, the criteria we sought in the articles are not presented clearly, or the information is scarce. Keeping in mind the broad nature of a scoping review and the aim of mapping the field, the author conducting the scoping study (EG) chose to err on the side of inclusion if the information in an article did not allow a decisive application of the criteria. While this strategy, combined with a single-rater approach, introduces a certain amount of selection bias, this bias was at least consistent between articles. We believe the approach provides a reasonable tradeoff between transparency and reproducibility of the study, and fulfilling the objectives of conducting a scoping review.

Other sources of bias were excluded by design. In particular, we conducted the consultation interviews after the literature sampling step, eliminating potential selection

bias arising from the interview results. While the strong dominance of articles treating tumour segmentation (cf. figure 2) may seem surprising, we believe that it is an accurate reflection of the research community's interest.

This review does not consider modalities such as projection radiography, CT, nuclear imaging or ultrasonography. It focuses instead on MR, which is used to address a larger variety of brain lesions, provides more detailed information and stands out among radiological techniques regarding the number of related publications.

We acknowledge self-critically that, due to the extensive nature of the task, the time taken, and the lessons learnt during the work, some epistemic drift occurred, taking our focus away from the first two RQs and towards the question of clinical applicability that appears to be underserved by the current literature. Thanks to our early decision to hold ourselves to account by writing a detailed protocol,[22] this drift stayed within reasonable boundaries and is well-documented, as we accounted for protocol modifications in the present work.

### Recommendations for future work
### RQ4 (b) What issues should be addressed in future studies to develop a tool that is suitable for clinical use?

To address the shortcomings revealed by our analysis, we propose a set of recommendations for future studies as well as avenues researchers might follow that promise, in our estimation, to advance the field in the direction of enabling clinical decision support.

The most important recommendation arises from the variability encountered in many aspects of the field. We propose a checklist (table 3) of items that should be reported in investigations of automatic brain lesion segmentation methods. We split the items into three levels of validation that we see as a potential path towards developing a clinical tool. Technical validation studies focus on developing the algorithm and evaluating its performance according to common criteria in the field. The preclinical validation focuses on the performance of an algorithm in a setting resembling clinical environment and a realistic use case. The final level, clinical validation, requires the tool to be compatible with hospital and radiology information systems and to be ready for the process of obtaining regulatory approval for clinical use if such approval has not been yet obtained. This checklist, especially the technical validation level, will facilitate replication as well as comparisons between methods and studies, both informal and in meta-analyses. If authors were to follow the checklist in future studies, this would be a step towards standardisation of reporting in the interest of advancing knowledge and promoting implementation as clinical tools.

With similar priority, we ask researchers who have developed or are working on automatic brain lesion segmentation algorithms to publish software implementations. The benefits and challenges with fully automated versus interactive preprocessing should be assessed in terms of segmentation accuracy and reproducibility for a given, clinically relevant task.

Few articles in our sample evaluated their algorithm on an independent data set on top of the challenge one. We encourage authors to endeavour to test their method on images from other sites and sources. We also encourage collaboration between the authors and independent researchers who may have access to annotated test images. A preprocessing and segmentation method that has an acceptable and consistent performance on images acquired from various sources should finally be tested in clinical conditions. Therefore we strongly advocate close collaboration between researchers and authors of well-performing methods with clinicians. In such a scenario, the acceptability of the method's performance to clinicians can be assessed in conjunction with its relevance for a given task. Moreover, an open dialogue between researchers and clinicians will help build an ABS system that meets the requirements for a clinically useful and usable tool.

Finally, efforts to define steps on the path towards designing and validating a clinically applicable ABS system should be made. We recommend consultation with stakeholders as a key element to verify the actual clinical needs and how to assess to what extent these needs are met by available research.

### CONCLUSIONS

This scoping study of automatic brain lesion segmentation on MR images shows a field growing at a rapid pace, an imbalance between proposed methods of which there are many and methods implemented for clinical application of which there are few, and a room for improvement of reporting practice with a view to enabling replication, method comparison and implementation. To promote this improvement, we propose a list of recommendations for future studies in the field. We identify knowledge gaps and potentially fruitful avenues for future research.

**Contributors** EG conducted the study and led the writing of the article. RAH was the main supervisor and consultant of the study progress and design choices. JS provided input on the study plan and methodology at all stages. IB-B provided guidance on the interpretation of the findings from a clinical perspective. All coauthors collaborated on manuscript composition and editing.

**Funding** JFS is supported by the Swedish Childhood Cancer Foundation (MT2018-0020) and the ATTRACT project funded by the EC under Grant agreement 777222.

**Competing interests** None declared.

**Patient consent for publication** Not required.

**Provenance and peer review** Not commissioned; externally peer reviewed.

**Data availability statement** Data are available in a public, open access repository. DOI: 10.6084/m9.figshare.13651235

terminology, drug names and drug dosages), and is not responsible for any error and/or omissions arising from translation and adaptation or otherwise.

**ORCID iDs**
Emilia Gryska http://orcid.org/0000-0002-7912-2232
Justin Schneiderman http://orcid.org/0000-0002-4441-2360
Isabella Björkman-Burtscher http://orcid.org/0000-0002-9023-3363
Rolf A Heckemann http://orcid.org/0000-0003-3582-3683

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
