## [Reviewer comments · BMJ Open]

ARTICLE DETAILS

TITLE (PROVISIONAL)	Automatic brain lesion segmentation on standard magnetic resonance images: a scoping review
AUTHORS	Gryska, Emilia; Schneiderman, Justin; Björkman-Burtscher, Isabella; Heckemann, Rolf

VERSION 1 – REVIEW

REVIEWER	marek czosnyka university of Cambridge, UK
REVIEW RETURNED	25-Aug-2020

GENERAL COMMENTS	Interesting and potentially useful review. Overall it is too long, and , forgive me, little bit boring. It lacks more clear presentation of references- like using factual tables rather than verbal presentation Legend of figures must be more descriptive
---

REVIEWER	Dr.M.Lavanya Saveetha School of Engineering, Saveetha Institute of Medical and Technical Sciences, Chennai, Tamilnadu, India
REVIEW RETURNED	21-Sep-2020

GENERAL COMMENTS	Kindly revise your paper based on the review comments. 1.The number of papers referred for review process is good and appreciable. 2.As only Magnetic resonance images are considered for automatic brain lesions why other types of images(X ray, CT, PET etc.,) are not considered reasons are to be justified. 3.Throughout the paper the general discussions of what reference articles have discussed are written in general but this paper is on segmentation so those methods are to be specified in the form of a table cited with reference papers and mention its advantages and limitations and parameters considered for validation of results. 4.Some suggestions on good algorithms can be given based on the results. 5.The image database referred can be mentioned. 6.The evaluation methods of segmentation are to be discussed. 7.The methodology used and results highlighted are to be included. 8.The future scope of segmentation methods to be specified. 9.The conclusion should be clear defining which is the best methodology.
---

REVIEWER	Ludovica Griffanti University of Oxford, UK
REVIEW RETURNED	01-Oct-2020

GENERAL COMMENTS	In this manuscript, the authors perform a scoping review on automatic brain lesion segmentation on MRI. They follow their published protocol, mapping the field in terms of types of lesions, algorithms used, evaluation protocols and clinical applicability. They then conclude with recommendations for future studies. To give a context to my review, I am not an expert of scoping reviews, my expertise is more on segmentation algorithms development and evaluation and I am also very interested in clinical translation. I have two major comments on this work. The first is that, while I agree with many of the recommendation for future work presented by the authors, I think that the whole review presents a large amount of concepts spanning a large amount of problems, not necessarily relevant to the final aim, which is to bridge the gap between research and clinical application. I understand that a scoping review is broad by design, but my overall comment is therefore to reframe the findings/sections on the review to focus on the final aim, so that the recommendations come as a logic conclusion of the observations (more details below). The second major point is that seems that the conclusion from this paper is to encourage studies to go from methods development, to validation, to replication, to implementation in clinical settings, to getting regulatory approval, all in one go (one study). This in my opinion is unfeasible and not the best approach to adopt. All these steps require different expertise, data, resources and timings. Provided that the scope of the algorithm is to be useful in clinic and that a clinical input has been received or taken into account, the first step is to develop an algorithm, which might be computationally intensive and not user-friendly, but the first limited evaluation works as a proof of concept. Then comes more extensive validation and replication, where more clinical collaboration is needed to get more datasets and validation. Only at a stage where there is more confidence that this method could work in clinical data is time to try and implement it and eventually get regulatory approval. I believe that studies at every stage should enable going ahead with the next step as smoothly as possible, so code sharing, data sharing and transparency of the methods are key, but only within the scope of that step of the development. My suggestion is therefore to re-structure/re-group the recommendations so that they are feasible to adopt by studies at different stages on the path of clinical application. E.g. code sharing for studies focused on algorithm development, more details/datasets/test-retest for studies providing validation on a larger scale, more clinical validation and PACS compatibility for methods that want to be ready for clinical translation etc. Here below are some additional comments and potential suggestions to consider. - The abstract, especially the objective, is very generic and should better clarify the aim of the review: "The standard practices of algorithm development and validation, however, have raised
--

	doubts.” doubt about what specifically? “examine clinical relevance of the published research.” very vague - As the authors say at some point, it is not necessarily relevant what preprocessing is applied to the images, especially because many use different modalities (sometimes specific for different lesions), which require different processing steps. What is more useful is if they need supervision, manual editing or intervention, or other steps that could represent a burden for clinical implementation. I would therefore encourage restructuring the section. - Related to the previous point, I agree that computing load is crucial, especially with new deep learning techniques, but is only mentioned in the recommendation and not in the rest of the text. In my opinion, one of the major gaps is that at the moment they require GPUs, which are not available in clinical settings. This is not discussed at all. I think that this should not discourage the use of advanced computing facilities to develop new methods, but it is part of the long path to clinical translation. As I said earlier, once the method is promising, then effort can be put to make it computationally lean and applicable in clinical settings. - Performance evaluation. While I recognise the amount of work that went into this paper, it is hard for me to see a way to evaluate in the same study tools to segment such a huge variety of lesions. It is true that some algorithms are applicable to different types of lesions, but the clinical need to segment lesions varies massively across types and consequently the most appropriate type of evaluation. For example brain tumours might be needed for surgical planning and need a very precise contour segmentation; for tumours a pattern analysis could help distinguishing types; MS lesions are usually counted, so evaluation and optimisation should be focused on lesion detection rather than on delineation; WMH are usually evaluated in terms of volume; strokes have different sub-regions (core/oedema etc). The authors should discuss and potentially give recommendation of which metrics are more appropriate for which task or encourage choosing the metrics according to the future clinical need. As a side note, the fact that the authors interviewed clinicians mostly in the field of brain tumours could have easily skewed the results towards this type of lesions and the needs in their field.
--	--

VERSION 1 – AUTHOR RESPONSE

Reviewer: 1

Reviewer Name: marek czosnyka
 Institution and Country: university of Cambridge, UK
 Competing interests: no competing interest

Please leave your comments for the authors below

R1C1: Interesting and potentially useful review.
 Overall it is too long, and , forgive me, little bit boring.

We appreciate the interest in our work and the recognition of its usefulness. We agree with the reviewer that the manuscript was long-winded. The revised manuscript is more succinct; in particular, we shortened *Stage 2: identifying relevant articles* and *Stage 3: study selection* in Section *Methods*. Where appropriate, results have been moved to tables.

The *Discussion* section (*Validation process and efficacy of automatic segmentation methods* and *Limitations and issues of the proposed methods, and grand challenges of the field*) have been carefully revised, which further resulted in shortening of the manuscript.

R1C2: It lacks more clear presentation of references- like using factual tables rather than verbal presentation

A spreadsheet listing the extracted information for each reference can be found in the supplementary material uploaded according to the journal's instructions (<https://figshare.com/s/9f6b51b4edcfd8fd9ef9> – please scroll down past the error message to the download link on the left side). We apologise if the material was not easily accessible during the review process. In the revised manuscript, the section References also includes all articles in the sample.

R1C3: Legend of figures must be more descriptive.

We agree with the reviewer's suggestion. We revised the figure legends in order to ensure the figures can be understood without reference to the main text:

"Figure 1: Flowchart of the article selection process from the result of querying three databases. For each stage, the number of articles selected is shown and numbers of excluded articles and reasons are given."

"Figure 2: Number of algorithms described in the included articles developed for and validated on particular lesion type. The distinction is made between multiple sclerosis (MS) lesions and other white matter lesions (WML) following the similar distinction present in the reviewed articles and lesion segmentation challenges. "Multiple lesion types" refers to algorithms that were evaluated on more than one type of lesion. "Other lesion types" types of lesions included focal cortical dysplasia, metastasis, traumatic brain injury, cerebral palsy, abscess, and necrosis."

"Figure 3: Number of articles published per year in the eligible sample with an indication of the number of articles that presented a segmentation method for a particular lesion type. We fit a linear function to the number of papers published each year from 2012 to estimate the development of the field in the upcoming years."

"Figure 4: The number of patients (n) whose images were used to validate the methods in the eligible articles."

"Figure 5: Distribution of the most common algorithms and models implemented in the lesion segmentation methods proposed by the articles eligible for the review (ANN – artificial neural networks, DT – decision trees, EM – expectation maximization, FC – fuzzy clustering, MM – mixture models, RF – random fields, SVM – support vector machines, THR – thresholding)."

Reviewer: 2

Reviewer Name: Dr.M.Lavanya
Institution and Country:
Saveetha School of Engineering,
Saveetha Institute of Medical and Technical Sciences,
Chennai, Tamilnadu, India
Competing interests: None Declared

Please leave your comments for the authors below
Kindly revise your paper based on the review comments.

Dear Authors,

R2C1: The number of papers referred for review process is good and appreciable.

We appreciate the encouraging comment.

R2C2: As only Magnetic resonance images are considered for automatic brain lesions why other types of images_(X ray, CT, PET etc.,) are not considered reasons are to be justified.-

Thank you for this valid question; we agree that the exclusion of other methods should be justified more clearly. Automatic brain lesion segmentation is by far most often applied to MR images. A larger variety of lesions and methods are furthermore addressed with MR, as compared to segmentation of brain lesions evaluated with CT. Conventional X-ray/projection radiography imaging generally is not used to evaluate brain lesions relevant for segmentation, PET, as a nuclear medicine modality, differs substantially from MR and also CT considering image resolution and image information. Considering these factors, we decided to limit the scope of this review to magnetic resonance imaging. In order to treat this valid point, we have amended the limitation section as follows: "This review does not consider modalities such as projection radiography, computed tomography, nuclear imaging, or ultrasonography. It focuses instead on MR, which is used to address a larger variety of brain lesions, provides more detailed information, and stands out among radiological techniques regarding the number of related publications."

R2C3: Throughout the paper the general discussions of what reference articles have discussed are written in general but this paper is on segmentation so those methods are to be specified in the form of a table cited with reference papers and mention its advantages and limitations and parameters considered for validation of results.

Agreed. Please refer also to our answer to comment R1C2. The information requested by the reviewer can be found in the supplementary material (<https://figshare.com/s/9f6b51b4edcfd8fd9ef9>). We apologise if the material was not easily accessible during the review process.

R2C4: Some suggestions on good algorithms can be given based on the results.

We partially agree with this point. A standard performance criterion is challenging to establish, especially considering the breadth of lesion types and variability in their presentations. We, therefore, refrain from attempting to rank methods herein. However, and thanks to this comment, we have been able to glean what could be considered procedural standards in brain lesion segmentation, at least through the preprocessing stage of analysis. Those are now more clearly summarized in Table 2. We furthermore address the challenge of evaluating methods by adding the following to the relevant part of the discussion "In the absence of widely used and agreed-upon criteria for performance evaluation, we have abstained from attempting to rank methods or to make specific recommendations."

R2C5: The image database referred can be mentioned.

Agreed. The majority of articles used publicly non-available or independently collected data sets. For those articles that used publicly available image databases that provide reference segmentations, the most common was BraTS [467]. We now present this information in Table 1 and give it more prominence by separating it from the information about MR sequences: instead of one *Materials* subsection under *Results*, the revised manuscript now has one subsection entitled *Image sources and databases* and one entitled *Sequences*.

R2C6: The evaluation methods of segmentation are to be discussed.

We agree. We restructured the first paragraph of the results section *Performance evaluation* for improved clarity. It now reads: “Comparison with the reference by overlap measures was by far the most commonly used criterion to evaluate the accuracy of the automatic segmentation. Among articles that specify how the reference segmentations had been generated (265) in publicly non-available data sets, most specify manual segmentation by one rater (85) or two raters (61). More than two raters contributed in 38 articles, and 33 articles used semiautomatic procedures for constructing the reference. When multiple raters delineated the region of interest, the reference segmentation was obtained through consensus of the raters or majority voting. Forty-five articles have also included inter- and intra-rater variability evaluation on data sets used in their studies.” We further have added the following to section *Validation process and efficacy of automatic segmentation methods*: “Three obstacles to fair comparative method assessments are paramount: the principal lack of ground truth in in vivo imaging, the enormous parameter space of acquisition settings that leads to variable feature presentations, and the lack of objectivity (as shown by inter-rater and intra-rater variability[478,482]) of reference segmentations.”

R2C7: The methodology used and results highlighted are to be included.

We used scoping review methodology as described by Arksey and O'Malley and other authors in cited papers [17-19]. We added the passage to the methods to ensure the methodology we used is clearly specified: “This study has been conducted according to a previously published protocol [22] and based on scoping review methods proposed by Arksey and O'Malley [17] and further developed by Levac et al. and Colquhoun et al. [18,19]. We also incorporate relevant parts of the PRISMA [20] and PRISMA-ScR [21] protocol for this type of review”. Since the review protocol has been previously published (cited paper [22]), we provide a somewhat abbreviated method description in Section *Method and Design*. We hope the reviewer agrees that duplicating information from the protocol paper would be redundant.

We sought to give prominence to key results (rapid growth of the field, imbalance between proposed methods and clinical applications, deficiencies of reporting practice) and have stated our key contributions in the abstract as well as in Sections *Results* and *Conclusions*. We have restructured the results and renamed the subsections to: “Image sources and databases”, “Sequences”, “Image processing steps, algorithms and computational theories” with two subsections “Preprocessing” and “Algorithms and computational theories”, “Performance evaluation”, and “Performance comparison”.

R2C8: The future scope of segmentation methods to be specified.

We anticipate continued growth of the field (cf. Figure 4). Regarding adoption in clinical practice, we have not found evidence of the required focussed development. Instead of making predictions, we therefore pointed out deficiencies in the reporting practice, along with possible remedies. In response to this comment and also to Reviewer 3 (R3C2), we have revised the relevant subsection under *Discussion*, entitled *Recommendations for future work*, and hope that the revision meets with the Reviewer's approval: “To address the shortcomings revealed by our analysis, we propose a set of recommendations for future studies as well as avenues researchers might follow that promise, in our estimation, to advance the field in the direction of enabling clinical decision support.”

R2C9: The conclusion should be clear defining which is the best methodology.

This point was also raised by the Reviewer in comment R2C4; please refer to our response there.

Reviewer: 3

Reviewer Name: Ludovica Griffanti
Institution and Country: University of Oxford, UK

Competing interests: None Declared

Please leave your comments for the authors below

In this manuscript, the authors perform a scoping review on automatic brain lesion segmentation on MRI. They follow their published protocol, mapping the field in terms of types of lesions, algorithms used, evaluation protocols and clinical applicability. They then conclude with recommendations for future studies.

To give a context to my review, I am not an expert of scoping reviews, my expertise is more on segmentation algorithms development and evaluation and I am also very interested in clinical translation.

I have two major comments on this work.

R3C1: The first is that, while I agree with many of the recommendation for future work presented by the authors, I think that the whole review presents a large amount of concepts spanning a large amount of problems, not necessarily relevant to the final aim, which is to bridge the gap between research and clinical application. I understand that a scoping review is broad by design, but my overall comment is therefore to reframe the findings/sections on the review to focus on the final aim, so that the recommendations come as a logic conclusion of the observations (more details below).

We thank the reviewer for the thorough appraisal of our work and constructive comments. We agree that the final aim of the reviewed research done in the field should be to produce a clinically useful tool, bridging the gap or even identifying steps to take on the way. Thus we formulated the purpose of our study as to “understand how the most prevalent methods (segmentation algorithms) and study designs (how they are validated) reflect the clinical applicability of methods described in the reviewed articles.” and to “identify issues, limitations, and grand challenges of the field, and suggest actions to bridge the gap between research and clinical practice.” The aims are also reflected, in our opinion, in our study design, where we established criteria to exclude purely technical studies, leaving only articles that, by design or claim, are clinically relevant. We excluded publications that did not provide segmentation evaluation; did not specify how the reference standard was obtained or the source of the processed images; did not provide the number of images used for evaluation, or evaluated the method on fewer than 10 unique scans. We considered technical accounts of a proposed algorithm to be outside the scope of our work.

In the eligible sample, we would have loved to extract more valid information that allows us to put the findings in the perspective of the clinical needs and environment. That perspective was, however, largely overlooked by the authors of the reviewed articles. This meant that we could not extract this information, only discuss the findings from the perspective of our aim. That being said, we have elected to restructure *Results* and rename the headings to better fit the posed research questions. The *Results* section now consists of the following subsections: “Image sources and databases”, “Sequences”, “Image processing steps, algorithms and computational theories”, “Performance evaluation”, and “Performance comparison”.

R3C2: The second major point is that seems that the conclusion from this paper is to encourage studies to go from methods development, to validation, to replication, to implementation in clinical settings, to getting regulatory approval, all in one go (one study). This in my opinion is unfeasible and not the best approach to adopt. All these steps require different expertise, data, resources and timings. Provided that the scope of the algorithm is to be useful in clinic and that a clinical input has been received or taken into account, the first step is to develop an algorithm, which might be computationally intensive and not user-friendly, but the first limited evaluation works as a proof of concept. Then comes more extensive validation and replication, where more clinical collaboration is needed to get more datasets and validation. Only at a stage where there is more confidence that this method could work in clinical data is time to try and implement it and eventually get regulatory approval.

I believe that studies at every stage should enable going ahead with the next step as smoothly as possible, so code sharing, data sharing and transparency of the methods are key, but only within the

scope of that step of the development.

My suggestion is therefore to re-structure/re-group the recommendations so that they are feasible to adopt by studies at different stages on the path of clinical application. E.g. code sharing for studies focused on algorithm development, more details/datasets/test-retest for studies providing validation on a larger scale, more clinical validation and PACS compatibility for methods that want to be ready for clinical translation etc.

Agreed. We revised the recommendations following the reviewer's valuable suggestions and believe that the new version is clearer, more appropriate, and more realistic: "We split the items into three levels of validation that we see as a potential path towards developing a clinical tool. Technical validation studies focus on developing the algorithm and evaluating its performance according to common criteria in the field. The pre-clinical validation focuses on the performance of an algorithm in a setting resembling clinical environment and a realistic use case. The final level, clinical validation, requires the tool to be compatible with hospital and radiology information systems and to be ready for the process of obtaining regulatory approval for clinical use if such approval has not been yet obtained."

We also agree with the reviewer that the original formulation of the recommendations was mismatched with what is feasible for method contributors. It was also mismatched with our own intention, as we do not actually expect the entire development chain to be covered in one study. Instead, we hope to convince contributors to see their role – and to define the scope of their work – as part of a development chain that does not end with the assessment of segmentation accuracy.

R3C3: The abstract, especially the objective, is very generic and should better clarify the aim of the review: "The standard practices of algorithm development and validation, however, have raised doubts." doubt about what specifically? "examine clinical relevance of the published research." very vague

We agree with the reviewer's point. We rephrased the abstract according to the suggestions. The passage now reads: "In this study, we map the field of automatic brain lesion segmentation to understand clinical applicability of prevalent methods and study designs, as well as challenges and limitations in the field."

R3C4: As the authors say at some point, it is not necessarily relevant what preprocessing is applied to the images, especially because many use different modalities (sometimes specific for different lesions), which require different processing steps. What is more useful is if they need supervision, manual editing or intervention, or other steps that could represent a burden for clinical implementation. I would therefore encourage restructuring the section.

We are thankful to the reviewer for this valuable suggestion. We agree that errors and failures in preprocessing and the potential burden that manual editing or intervention poses for clinical application deserve recognition in our discussion. We revised the *Variable design of automatic segmentation* methods section and included the suggestion from the reviewer: "Another implementation issue arises from the necessity of visual supervision or manual corrections of the preprocessing that was indicated in 15 articles. Any requirement of user interaction in the processing chain entails disadvantages: it impedes implementation by increasing the complexity of the technical integration, and, once the tool is implemented, permanently burdens staff with an additional task. Any potential benefit of user interaction (e.g. increased robustness) has to be weighed carefully against these costs."

R3C5: Related to the previous point, I agree that computing load is crucial, especially with new deep learning techniques, but is only mentioned in the recommendation and not in the rest of the text. In my opinion, one of the major gaps is that at the moment they require GPUs, which are not available in clinical settings. This is not discussed at all. I think that this should not discourage the use of advanced computing facilities to develop new methods, but it is part of the long path to clinical translation. As I said earlier, once the method is promising, then effort can be put to make it computationally lean and applicable in clinical settings.

We agree with the reviewer that the limited availability of GPUs in a clinical setting adds to the implementation gap. We followed the suggestion and now discuss the issue in the section *Relevance for research and clinical practice* as follows: “Another finding that potentially contributes to the gap between research and clinical practice is the widespread use of deep-learning algorithms for the lesion segmentation task in the research setting. The practical efficiency of the algorithms depends on the availability of graphical processing units (GPUs). Implementing a deep-learning based algorithm in a clinical setting requires dedicating resources for purchasing suitable hardware and integrating it with radiological workflow, information, and storage systems.”

R3C6: Performance evaluation. While I recognise the amount of work that went into this paper, it is hard for me to see a way to evaluate in the same study tools to segment such a huge variety of lesions.

It is true that the range of lesion types that have been tackled with automatic segmentation methods is too broad for a meta-analysis or other type of systematic literature review with a quantitative intent. Such a review would need to focus more narrowly. We expected this and chose scoping methodology for this reason, partly with a view to informing our own future efforts, but (perhaps obviously) in the hope that the review findings would also be of value to other researchers in the field. We also followed a clinical perspective: lesions on MR are signal changes based on pathophysiological processes that are similar for many different diseases and thus often not per se pathognomonic but subject to differential diagnosis. Thus, we believe that this scoping review needs to embrace different pathologies also to encourage broader future studies. We hope the reviewer agrees that we have achieved this and that our paper does indeed provide valuable guidance for future review studies, as well as for method development work and reporting thereof.

R3C7: It is true that some algorithms are applicable to different types of lesions, but the clinical need to segment lesions varies massively across types and consequently the most appropriate type of evaluation. For example brain tumours might be needed for surgical planning and need a very precise contour segmentation; for tumours a pattern analysis could help distinguishing types; MS lesions are usually counted, so evaluation and optimisation should be focused on lesion detection rather than on delineation; WMH are usually evaluated in terms of volume; strokes have different sub-regions (core/oedema etc). The authors should discuss and potentially give recommendation of which metrics are more appropriate for which task or encourage choosing the metrics according to the future clinical need.

As also reflected in our comment above, we agree with the reviewer’s position that it is important to match a segmentation evaluation method with a clinical need the method will fulfil. We expanded the discussion of evaluation metrics accordingly, pointing out the importance of a proper selection of the validation metrics: “Various metrics may be used to either optimize the method or evaluate its accuracy for a particular clinical question, lesion type, and size.” The information we extracted in this study does not, however, provide a basis for providing more detailed recommendations, as this is outside our already broad scope for the paper.

R3C8: As a side note, the fact that the authors interviewed clinicians mostly in the field of brain tumours could have easily skewed the results towards this type of lesions and the needs in their field.

It is true that our results suggest that among the possible applications of automatic brain lesion segmentation, brain tumours receive the most attention in the literature. This could be an accurate reflection of reality or a result of subconscious bias on our part. However, while the reviewer's concern is legitimate, we consider it unlikely that the interviews have led to substantial bias as the inclusion criteria did not concern any particular lesion type.

VERSION 2 – REVIEW

REVIEWER	Dr.M.Lavanya Saveetha school of engineering, Chennai, TamilNadu, India
-----------------	---

REVIEW RETURNED	12-Dec-2020
GENERAL COMMENTS	 1. Sample lung lesion segmentation could be included. 2. Graphical flowchart to be included to understand the work flow.

VERSION 2 – AUTHOR RESPONSE

Reviewer: 2

Dr. M Lavanya, Saveetha University, Saveetha school of engineering

Comment 1. Sample lung lesion segmentation could be included.

Response:

Imaging of lung lesions and segmentation of these is a major research field that mainly focuses on computed tomography and clinical applications are available. The scope of our review was defined at the protocol stage. To provide a comprehensive and rigorous review of the field of brain lesion segmentation based on magnetic resonance (MR) images, we strictly followed the protocol and did not mix different modalities and body parts. Adding a sample of lung lesion segmentation is thus beyond the scope of this work.

Comment 2. Graphical flowchart to be included to understand the work flow.

Response:

We agree that the visualization of the workflow is important and thus included a flowchart of article selection (Figure 1, Flowchart of the article selection process from the result of querying three databases. For each stage, the number of articles selected is shown and numbers of excluded articles and reasons are given.). Fig. 1 includes the methods Stages 2 and 3 and Stage 4 is included as supplementary information. Stage 5 is then presented in the Results tables and text. It would therefore not aid the reader's understanding to include more information in the flowchart.